# Assessment of long-lasting insecticide nets coverage, utilization, and associated factors among households in malaria elimination districts of Arsi Zone, Oromia Region, Ethiopia: A cross-sectional study

**Olana Yadate Hunde**[1], **Haimanot Ewnetu Hailu**[1]*, **Jimmawork Wondimu**[1],
**Belachew Dinku**[1], **Wegene Ewnetu**[2]

1 Department of Public Health, St. Paul's Hospital Millennium Medical College, Addis Ababa, Ethiopia,
2 Saint Peter's Specialized Hospital, Addis Ababa, Ethiopia

☯ These authors contributed equally to this work.
* metiewnetu@gmail.com

## Abstract

### Background

Malaria is a major global public health problem, with a particular burden of disease in sub-Saharan Africa including Ethiopia. Access to Long Lasting Insecticide Nets (LLINs) for at-risk populations, ensuring its appropriate utilization and identifying the barriers are important for malaria prevention, control and elimination. This study aimed to assess coverage, utilization and associated factors of Long Lasting Insecticide Nets (LLINs) among households in the Arsi Zone of Oromia Region, Ethiopia.

### Methods

Community-based cross-sectional study was conducted from October to December 2021. Multi-stage sampling technique was used to recruit 1250 households from five districts out of 21 Malarious districts in the Zone. Proportional allocations of households were done in each sampled kebeles and simple random sampling was used to draw the study participants. Data were collected by trained data collectors using a pre-tested structured questionnaire and observation. The collected data were exported to and analyzed using SPSS version 23. Variables with a p-value below 0.2 at bivariable logistic regression analysis were entered into the multivariable logistic regression model. We presented findings using an adjusted odds ratio with 95%CI at a p-value of less than 0.05.

### Results

Out of the total of 1250 households 99.5% of surveyed owned LLINs and 27.1% of them had slept under the net the night before the survey. The factors associated with LLIN usage included being in the age range of 40 to 49 years (AOR; 1.82, 95%CI 1.01–3.25), preference

**Data Availability Statement:** Full data is available on figshare:DOI: 10.6084/m9.figshare.24189888.

**Funding:** OYH received the award from CDC Ethiopia under The President's Malaria Initiative (PMI) No: The funders had no role in study design, data collection and analysis, decision to publish, or preparation of the manuscript.

**Competing interests:** The authors have declared that no competing interests exist.

**Abbreviations:** CDC, Center For Disease Control; DHS, Demographic Health Survey; FMOH, Federal Ministry Of Health; GTS, Global Technical Strategy; HHs, House Holds; IRS, Indoor Residual Spray; ITNs, Insecticide Treated Nets; LLINs, Long Lasting Insecticide Nets; MIS, Malaria Indicator Survey; NMSP, National Malaria Strategic Plan; RBM, Roll Back Malaria; SPHMMC, Saint Paul's Hospital Millennium Medical College; SPSS, Statistical Package For Social Science; UNICEF, United Nations Children's Fund; WHO, World Health Organization.

for conical-shaped LLINs (AOR = 2.36; 95% CI: 1.33–4.18), not believing LLINs expired within 6 months (AOR 3.75, 95% CI 2.31–6.09), reporting a mosquito bite as a mode of malaria transmission (AOR = 2.46; 95%CI: 1.01–5.98), employed (AOR = 9.0; 95%CI: (4.22–20.02) and type of sleeping bed (AOR =: 17.4; 95% CI, 11.74–26.03). On the other hand, households with two and above sleeping rooms were less likely to use LLINs (AOR = 0.46; (95% CI: 0.23–0.88).

## Conclusion

Even though the ownership of Long Lasting Insecticidal Nets was high, the actual utilization was very low. Promoting the usage of LLINs utilization among those at most risk, through intensified health education activities will be helpful.

## Background

Malaria is caused by the protozoan parasite Plasmodium, which is transmitted by female Anopheles mosquitoes. Plasmodium falciparum remains the most dangerous and is responsible for the majority of malaria-related deaths than other species like P. vivax, P. malariae, P. ovale, and P. knowlesi [1]. Malaria continues to cause unacceptably high levels of morbidity and mortality. According to WHO report, there were an estimated 229,000 cases and 409,000 deaths globally in 2019 [2]. About 90% of all malaria deaths in the world today occur in Africa south of the Sahara. An estimated one million people in Africa die from malaria each year and most of these are children under 5 years old [3]. In addition to health impacts; malaria causes; a significant impairment to social and economic development; loss of workforce time both of the sick and the family members, who provide care, depletion of income as well as school absenteeism [4].

In Ethiopia, malaria is one of the major public health challenges with an estimated 68% (52 million) of the population is at risk of contracting malaria. Malaria transmission is seasonal and predominantly unstable, with frequent and often large-scale epidemics [5,6]. The main malaria parasites are P. falciparum and P. vivax, accounting for 60% and 40% of all cases, respectively. Anopheles arabiensis is the main vector and Anopheles pharoensis is also widely distributed in the country and is considered to play a secondary role in malaria transmission [6,7].

The government of Ethiopia is exerting a huge work force to fight against malaria starting during 2004. Accordingly, significant reductions in malaria mortality and morbidity have been achieved in the country in the last decades [8]. Currently; National Malaria Strategic Plan has been developed in alignment with the Global Technical Strategy (GTS) for Malaria elimination, which set milestones for 2030. The National Malaria Strategic Plan (NMSP) sets for achieving more than 75% reduction in malaria related morbidity and mortality by the year 2025 from the baseline of 2015 and malaria elimination targets by 2030 through different key strategic interventions [8].

The main strategies and interventions for malaria prevention, control and elimination are strengthening malaria vector control measures which help reducing-human-vector contact and controlling adult mosquito through distribution of Long Lasting Insecticide Nets (LLINs) and ensuring its appropriate utilization and targeted Indoor Residual Spray (IRS) [7].

Thus, appropriate utilization of LLINs and identifying the barriers or enablers are important for responding to and scale-up strategies to keep up the momentum of malaria

prevention, control and elimination strategies. Therefore, this study aims to assess LLINs coverage, utilization and associated factors of among households in malaria elimination districts of Arsi Zone, Oromia Region, Ethiopia.

## Methods and materials

A community based cross-sectional study was conducted in districts of Arsi Zone of Oromia Region, Ethiopia. Multi-stage sampling followed by simple random sampling technique was used to recruit 1250 households from five districts out of 21 Malarious districts in the Zone from October to December 2021. The sample size was calculated by using single population proportion formula using proportion of LLINs utilization of 60.6% from similar study [9]. Design effect of 2 and 10% nonresponse rate were used for final sample size.

A structured questionnaire was administered to identify socio-demographic, LLINs availability and status, LLIN use, and observation was made to inspection condition and handling of the LLINs. The tool was translated to Afan Oromo (local language) and back translation was done to check consistency. Then, women and/or head of HHs were interviewed to obtain respective information. Direct observation of the LLINs was made to verify their condition and where it was placed in the house. Each household was labeled by unique identification number during interviews and the authors had access to information that could identify individual participants during or after data collection.

The data collected was checked for completeness and entered into Epidata for data cleaning. Then, it exported to SPSS version 23 for data analysis. Frequencies and proportions were calculated for socio-demographic characteristics, and knowledge, perception and practices related to malaria transmission and prevention. Utilization of LLIN was estimated based on whether households have used at least one LLIN the night before the assessment took place.

A binary logistic regression analysis was used to examine the determinants of LLIN utilization and variables with a p-value below 0.2 during bivariate analysis was entered into the multivariable logistic regression model. A p-value of less than 0.05 in multivariable logistic regression analysis was used to declare a statistical significance. Both crude and adjusted odds ratios were presented with a 95% confidence interval. Cases with missing data were eliminated in the multivariate logistic regression analyses.

## Operational definitions

- **HH fully covered by LLINs:** A household with LLINs for each and every sleeping area/beds and observed by enumerators during data collection [10].

- **LLINs ownership:** HHs with one or more LLINs per household [10].

- **Proper utilization:** Refers to HHs that owned LLINs in which one or more members of the HH slept under a net, confirmed through observation by enumerators during the early morning preceding this study [11].

- **Access:** The proportion of the population that could sleep under an LLIN if each LLIN in the household was used by up to two people

- **LLINs:** Are nets that are treated at factory level by a process that binds or incorporates insecticide into the fibers. They are designed to maintain their biological efficacy against vector mosquitoes for at least 3 years [12].

## Results

### Socio-demographic characteristics of respondents

A total of 1250 households were included in this study. Of the total households included in the study, 1005 (80.4%) of the respondents were females. The median age of the respondent was 37.8 years. Family size ranged from 1 to 8, with a mean of 4.9 persons per household. The surveyed households had a total of 6134 family members; of which 762 (12.4%) were children under 5 years and 209 (3.4%) were pregnant women. Most of the respondents 873 (69.8%) were farmers and 568 (45.4%) of them had not attended formal education.

### Malaria related awareness of respondents

Almost all of the participants had heard about malaria as it is the health problem and mention that mosquito bites were the cause of malaria. The majority of participants 1220 (97.6%) believed that LLINs is used to prevent malaria (Table 1).

### LLINs coverage and utilization

All visited households owned at least one LLIN. However, out of the 1250 households that owned LLINs, 349 (27.1%) had slept under it the night prior to the survey. LLINs are mainly utilized by all family members in more than ¾th (81.8%) of the households, while children under 5 years of age or pregnant mothers are given priority in only 135 (10.8%) and 93 (7.4%) households respectively. Regarding periods of LLINs use, 36.3% of respondents used it during rainy season while 23.8% of the study participants used LLINs throughout the year. Respondents underscored the role of weather as one of the reasons for inconsistent LLIN use. However, only 8.1% use LLIN during hot weather because it is uncomfortable and they frequently use it during rainy seasons as malaria transmission is higher during rainy session, and the temperature is relatively low according to their reports. Some households were using LLINs for unintended purposes like rapping over mattresses to protect from bugs, for carrying grain, for

**Table 1. Awareness of households on malaria and long-lasting insecticide nets in Arsi Zone of Oromia Region, Ethiopia, 2021.**

| Variables | Categories | Frequency | Percentage |
|---|---|---|---|
| Heard about malaria | Yes | 1244 | 99.5 |
| | No | 6 | 0.5 |
| Malaria transmission | Mosquito bite | 1185 | 94.8 |
| | Other modes | 65 | 5.2 |
| Preventive measure | LLINs | 1220 | 97.6 |
| | IRS | 23 | 1.8 |
| | Drainage stagnant water and others | 7 | 0.6 |
| Mosquito biting time | All the time | 221 | 17.7 |
| | At night time | 990 | 79.2 |
| | Day time | 39 | 3.1 |
| Educational message on LLINs | Yes | 1227 | 98.2 |
| | No | 23 | 1.8 |
| Source of information | HEW | 1033 | 82.6 |
| | Media | 199 | 16.0 |
| | Others | 18 | 1.4 |
| Use of LLINs | Yes | 1208 | 96.6 |
| | No | 42 | 3.4 |

**Table 2. Long lasting insecticidal nets utilization in Arsi Zone of Oromia Region, Ethiopia, 2021.**

| Variables | Categories | Frequency | Percentage |
|---|---|---|---|
| Were LLINs used last night | Yes | 349 | 27.9 |
| | No | 901 | 72.1 |
| Reason for not using | No malaria now | 532 | 42.6 |
| | For other purpose | 324 | 25.9 |
| | Too hot | 28 | 2.2 |
| | Attract other insects | 17 | 1.4 |
| Priority to sleep under LLIN | Under five children | 135 | 10.8 |
| | Pregnant women | 93 | 7.4 |
| | All family | 1022 | 81.8 |
| Periods to use LLINs | All the year | 298 | 23.8 |
| | Dry season | 101 | 8.1 |
| | Rainy season | 454 | 36.3 |
| Condition of LLINs | Good | 560 | 44.8 |
| | Poor | 690 | 55.2 |
| Sleeping place | Foam mattress | 646 | 51.7 |
| | Grass mattress | 604 | 48.3 |
| Where it found | Inside | 695 | 55.6 |
| | Outside | 305 | 24.4 |
| | Discarded | 250 | 20.0 |
| Walls of the room status | Concrete | 585 | 46.8 |
| | Mud with wood frame | 665 | 53.2 |

spreading grains on the sunshine, using as curtain and other similar purposes were observed during the assessment (Table 2 and Fig 1).

## Factors associated with LLINs utilization

Variables with a p ≤ 0.2 during the bivariate analysis were included in the final model. In the multivariable analysis, age of respondents, residence, occupational status of head of household, perception about LLINs expiry date, awareness on the mode of malaria transmission, number of sleeping rooms, type of sleeping beds, and preference for conical-shaped LLINs were significantly associated with LLINs utilization. Households in age group of 40 to 49 years were 1.82 times more likely to use LLINs as compared to under 30 years (AOR; 1.82, 95%CI 1.01–3.25). The odds of using LLINs among the employed were 9 times (AOR = 9.0; 95%CI: (4.22–20.02) compared to the merchants. The odds of using LLINs among households of urban residents were 4.09 times (AOR = 4.09; 95%CI: 2.53–6.62) compared to rural residents. Accordingly, the odds of using LLINs among respondents who said malaria was transmitted through mosquito bite was 2.46 times (AOR = 2.46; 95%CI: 1.01–5.98) higher than those who mention another mode such as shortage of food. Additionally, the odds of LLINs use were 3.75 times higher among households that did not perceive LLINs expired within 6 months (AOR 3.75, 95% CI 2.31–6.09) than those who perceive LLINs expired within 6 months. On the other hand, households with two and above sleeping rooms were 54% less likely to use LLINs than those with one sleeping room (AOR = 0.46; (95% CI: 0.23–0.88). The odds of LLIN use among households who use foam mattress for sleeping was 17.4 times (AOR = 17.4; (95% CI: 11.74–26.03) higher as compared with those who use grass mattresses for sleeping. Similarly, respondents who preferred conical-shaped LLINs were 2.36 times more likely to use LLINs than those who preferred rectangular shaped (AOR = 2.36; 95% CI: 1.33–4.18) (Table 3).

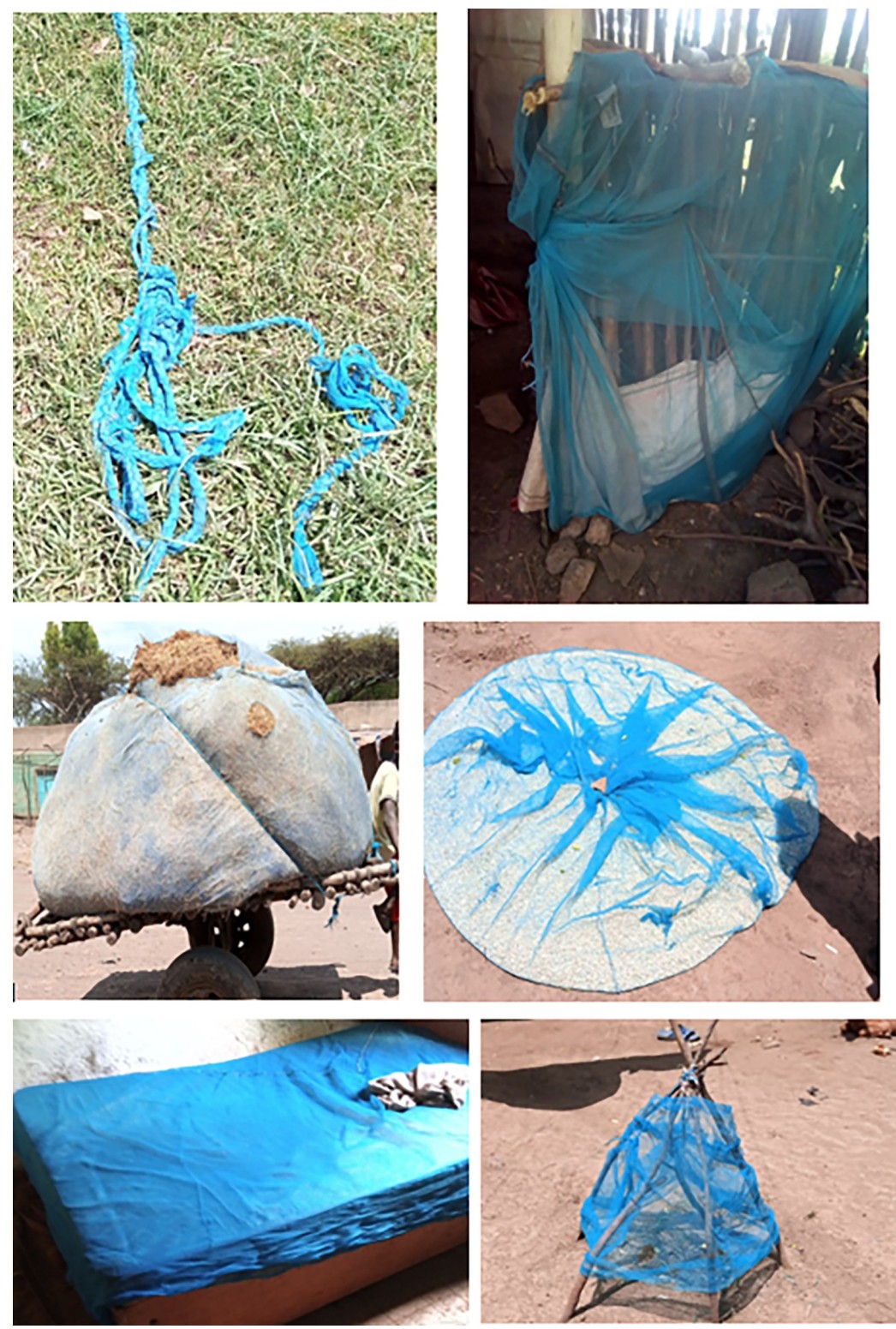

**Fig 1. Misuse and inappropriate use of LLINs.**

**Table 3. Multivariable analysis for factors associated with the use of LLINs in Arsi Zone of Oromia Region, Ethiopia, 2021.**

| Variables | LLIN utilization | | COR (95%CI) | AOR (95%CI) | P-value |
|---|---|---|---|---|---|
| | Yes | No | | | |
| Age of respondents <30 | | | | | |
| 30–39 | 49 | 179 | 1 | 1 | .816 |
| 40–49 | 149 | 437 | 1.25(0.86–1.79) | 1.05(0.66–1.70) | **.043** |
| >50 | 95 | 182 | 1.91(1.27–2.85) | 1.82(1.01–3.25) | .557 |
| | 56 | 103 | 1.98(1.26–3.13) | 1.21(0.63–2.33) | |
| Area of residents Rural | | | | | |
| Urban | 259 | 591 | 1 | 1 | **.001** |
| | 90 | 310 | 1.51(1.14–1.99) | 4.09(2.53–6.62) | |
| Educational status of HH | | | | | |
| No formal education | 191 | 377 | 1.29(0.97–1.72) | 0.85(0.53–1.32) | .519 |
| Primary | 54 | 259 | 0.53(0.36–0.77) | 0.62(0.36–1.05) | .077 |
| Secondary and above | 104 | 265 | 1 | 1 | |
| Housewife education | | | | | |
| No formal education | 270 | 581 | 1 | 1 | .742 |
| Primary | 29 | 117 | 0.53(0.34–0.82) | 0.92(0.52–1.59) | .278 |
| Secondary and above | 50 | 203 | 0.53(0.37–0.75) | 0.73(0.42–1.28) | |
| Occupational of HH head Daily laborer | | | | | |
| Gov't Employee | 13 | 55 | 1.47(0.67–3.22) | 6.0(2.16–16.68) | **.001** |
| Farmer | 55 | 98 | 3.49(1.92–6.35) | 9.0(4.22–20.02) | **.001** |
| Merchant | 263 | 636 | 2.57(1.53–4.32) | 3.6(1.89–6.88) | **.001** |
| | 18 | 118 | 1 | 1 | |
| LLINs expire in 6 moths | | | | | |
| Yes | 31 | 178 | 1 | 1 | **.001** |
| No | 318 | 723 | 2.52(1.68–3.78) | 3.75(2.31–6.09) | |
| Heard about malaria | | | | | |
| Yes | 339 | 862 | 1 | 1 | .066 |
| No | 10 | 39 | 0.65(0.32–1.32) | 0.45(0.19–1.29) | |
| Malaria transmission | | | | | |
| Mosquito bite | 341 | 8 | 2.88(1.36–6.10) | 2.46(1.01–5.98) | **.048** |
| Another mode | 844 | 57 | 1 | 1 | |
| Had information on LLINs | | | | | |
| Yes | 336 | 13 | 0.29(0.12–0.66) | 0.49(0.19–1.28) | .149 |
| No | 891 | 10 | 1 | 1 | |
| Sleeping room | | | | | |
| Less than two | 65 | 205 | 1 | 1 | **.021** |
| Two & above | 284 | 696 | 1.28(0.94–1.76) | 0.46(0.23–0.88) | |
| Type of sleeping bed | | | | | |
| Foam **Mattress** | 293 | 56 | 8.12(5.92–11.36) | 17.4(11.74–26.03) | **0.001** |
| Grass mattress | 353 | 548 | 1 | 1 | |
| Shape preferred | | | | | |
| Rectangular | 65 | 210 | 1.50(0.91–2.52) | 0.000 | **.003** |
| Conical | 259 | 569 | 2.22(1.41–3.500) | 2.36(1.33–4.18) | |
| Others | 25 | 122 | 1 | 1 | |

## Discussion

This study has identified that the proportion of households that own LLINs is very high however the utilization was very low. It is also identified that age, occupation, perception of LLINs expiry date, shape of LLINs are among the factors that are associated with the utilization of LLINs. Ethiopian Federal Ministry of Health works substantially to increase ownership of LLINs in each malaria-endemic areas, but, many barriers have been indicated in terms of proper utilization of LLINs. In this study, there was a high LLIN coverage (99.5%). Even

though almost all households received LLINs during the campaign, only 27.1% of them were seen to have hung during the study. This study finding in line with those conducted in Southwest, Ethiopia. However, LLIN utilization in this study was higher than in a study conducted in other parts of Ethiopia [13–15]. The possible explanation is to meet the target of universal access, WHO recommends that one LLIN be distributed to at least every two people at risk for malaria [16].

The benefits of LLIN use were recognized mainly as malaria prevention. In this study, all visited households owned at least one LLIN. However, out of the 1250 households that owned LLINs, 27.1% (349) had slept under the prior night. This study finding is lower than utilization by people living in malaria-endemic areas in Africa and the overall bed net utilization in Ethiopia [16–18]. Most households were using LLINs for unintended purposes like rapping over mattresses to protect from bugs, for grain carrying, spreading grains in the sunshine, using them as curtains, and other similar purposes were observed during this study. The basic reason for low utilization could be due to low awareness of consistent use of LLINs considering malaria transmission is mainly during rainy season. Additionally; there is a perception that LLIN itself will expire after six months of distribution and expectation for the new LLIN supply every year.

Age was significantly associated with LLIN usage, where households headed by those aged 40 to 49 years were the highest of other age groups having slept under LLINs during the night preceding the survey. This is in line with the findings of several previous studies elsewhere [14,17,18] and is in accordance to similar study in Southwest Ethiopia [14]. This could be due the fact that older adults may have a higher awareness of their health needs and family responsibility.

Our findings indicate that LLINs use was higher among urban dwellers. This finding is similar with the study in Ethiopia [19] where the same higher use of LLIN in urban areas was reported. The possible reasons might be due to increased education, potentially larger and more diverse social networks perhaps due to greater population density or a more progressive attitude among urban dwellers leading to earlier adoption with strategies. Our results also identified that the odds of using LLINs among government employee were 9 folds higher than households with other occupation groups. The possible explanation may be individuals with government or private employee had more awareness from different sources on the risk of malaria, mode of transmission and preventive measures, thus they are more likely to sleep under a net than other individuals [14].

There is still a misconception regarding awareness about causes and transmission of malaria in this community. Accordingly, the odds of using LLINs among respondents who said malaria transmitted through mosquito bite was 2.46 times higher than those who reported other modes of transmission. Other study conducted in Ethiopia revealed that the role of mosquitoes in malaria transmission was recognized only by 67% of respondents [14]. The possible reasons might be due to low educational level, less exposure or access to health information. Universal access to LLINs is best achieved by free mass distribution campaigns every 3 years or less [16]. The present study found that households that did not perceive LLINs expired within 6 months were less like to use the LLIN, which is concordant with the finding of the study in Northwest Ethiopia [20]. The possible explanations might be less awareness of users as the chemical impregnation of nets no is longer functional after six months.

Households with number of room or separate sleeping beds greater than two and above were less likely to utilize LLINs than those with number of beds or separate sleeping space of one room. The study from Nigeria and Easter Rwanda was in agreement with this study [13,21], but a study conducted in Kersa and Gamo Gofa Zone, Ethiopia were contrary to our result [15,22]. On the other hand, similar to this study, family size was also an important predictor for LLIN use in Southwest Ethiopia where households with a family size one to three

were 1.7 times more likely to use LLINs than those with seven or more [14]. The reason for this could be increased space and sleeping rooms, makes it harder even though they have an adequate number of LLINs for the size of the family.

The shape of LLINs had a considerable influence on utilization. In this study, conical shaped LLINs were significantly associated with LLINs usage. In a similar way, in a study conducted in Limmu Seka District of Oromia Region of Ethiopia, conical LLINs were more likely have been used the previous night when compared with rectangular shape [14]. Being rectangular in shape, it is possible that there were difficulties in hanging the LLINs in the circle-shaped huts, which may discourage their use and that have been reported as factors affecting the use of bed nets elsewhere [20,23]. In our justification, the preference might be due to the fact that conical LLINs use a single point for hanging. Additionally, Health extension workers in our study suggested that greater persistent utilization of LLINs may be achieved through wider distribution of circular or conical nets instead of rectangular nets, that might be due to rectangular shape is difficult to hang.

Similarly, type of sleeping bed was one of the factors associated with bed net utilization in this study. Using Foam mattress for sleeping on bed was significantly associated with LLIN usage than households using grass mattress in our study. The finding is slightly in line with a study reported from Kenya that sleeping on the floor decreases net utilization [24]. Similar study also revealed that, the net use of residents who slept on beds in bedrooms was greater than that in the other arrangements and the net use of residents on beds in non-bed rooms was greater than that of those without beds in non-bedrooms [25]. The possible reason might be that those who had foam mattress on their bed helps to tucking the nets over their bed and make them comfortable to use. Additionally, grass mattress mostly used without bed or on the floor out of bed rooms which might make it to use LLIN on such sleeping areas.

Despite the significant contribution of the findings, the limitations of this study include the very nature of the design which makes it hard to establish the temporal relationship between the factors and LLIN utilization. Secondly, since this is based on the report of the respondents, there is a possibility of recall bias and social desirability bias regarding the utilization of the LLINs. Additionally, other multiple factors and the interrelationship between these factors might affect LLINs utilization and this in turn might introduced a classification bias. Also, the findings should be interpreted by taking into consideration that the study did not assess the prior level of community's exposure to Social and Behaviour Change programs.

## Conclusions

This study identified that Long Lasting Insecticidal Net coverage was high, but the actual utilization was low. Age of respondents, residence, occupational status head of household, misperception of LLINs expiry date, awareness on mode of malaria transmission, number of sleeping room, shape preference of LLINs, type of sleeping mattress were identified as associated factors with LLINs utilization.

This finding will provide some insight on factors that should be taken into consideration during the promotion of utilization of LLINs through health education, mass media campaigns and continuous follow-up. It might be hard to generalize this finding to external population as this is one of the areas selected for malaria elimination with a number of interventions.

## Acknowledgments

The authors would like to thank the Public health Department of SPHMMC and CDC Ethiopia, the Oromia Regional health bureau, the Arsi zone health office, all respected district health offices, all data collectors, supervisors, and study participants.

## Author Contributions

**Conceptualization:** Olana Yadate Hunde.

**Formal analysis:** Olana Yadate Hunde, Haimanot Ewnetu Hailu, Jimmawork Wondimu, Belachew Dinku, Wegene Ewnetu.

**Funding acquisition:** Olana Yadate Hunde.

**Investigation:** Olana Yadate Hunde.

**Methodology:** Olana Yadate Hunde, Haimanot Ewnetu Hailu, Jimmawork Wondimu, Belachew Dinku, Wegene Ewnetu.

**Project administration:** Olana Yadate Hunde, Jimmawork Wondimu.

**Software:** Olana Yadate Hunde.

**Supervision:** Haimanot Ewnetu Hailu, Jimmawork Wondimu, Belachew Dinku, Wegene Ewnetu.

**Writing – original draft:** Olana Yadate Hunde.

**Writing – review & editing:** Haimanot Ewnetu Hailu, Jimmawork Wondimu, Belachew Dinku, Wegene Ewnetu.

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
