## [Decision Letter · Decision Letter 0]

23 Aug 2023

PONE-D-23-19534Assessment of Long-Lasting Insecticide Nets Coverage, Utilization, And Associated Factors Among Households in Malaria Elimination Districts of Arsi Zone, Oromia Region, EthiopiaPLOS ONE

Dear Dr. Hailu,

Thank you for submitting your manuscript to PLOS ONE. After careful consideration, we feel that it has merit but does not fully meet PLOS ONE’s publication criteria as it currently stands. Therefore, we invite you to submit a revised version of the manuscript that addresses the points raised during the review process.

We look forward to receiving your revised manuscript.

Kind regards,

Bosco Bekiita Agaba, PhD

Academic Editor

PLOS ONE

3. We note that Figure 1 in your submission contain copyrighted images. All PLOS content is published under the Creative Commons Attribution License (CC BY 4.0), which means that the manuscript, images, and Supporting Information files will be freely available online, and any third party is permitted to access, download, copy, distribute, and use these materials in any way, even commercially, with proper attribution. For more information, see our copyright guidelines: http://journals.plos.org/plosone/s/licenses-and-copyright.

Additional Editor Comments:

Reviewer 2

Abstract and the background chapter are well written. A Few comments on the methods;

Methods:

”……. The sample size was calculated by using single population proportion formula using proportion of LLINs utilization of 61.6% from similar study” … Please include this refence

“A structured questionnaire was administered to identify socio-demographic, LLINs availability and status, LLIN use, and observation was made to inspection condition and handling of the LLINs” there are multiple factors that can potentially affect LLINs utilization, did this tool capture exhaustively capture all these factors/variables? Would it be good idea to indicate how this could have affected your stud and if possible discuss this as a limitation?

“A binary logistic regression analysis was used to examine the determinants of LLIN utilization

and variables with a p-value below 0.2 during bivariate analysis was entered into the

multivariable logistic regression model” ……. Why was this cutoff set at 0.2? doesn’t setting a much “relaxed”/ less stringent cut off at variable selection potentially retain many unnecessary variables progressing to the final models which essentially could have been dropped

Results

“The median age of the respondent was 37.8 years…” Whats the age mostly affected by malaria in Ethiopia? Is it the under 5?

Reviewers' comments:

Reviewer's Responses to Questions

**Comments to the Author**

1. Is the manuscript technically sound, and do the data support the conclusions?

Reviewer #1: Yes

2. Has the statistical analysis been performed appropriately and rigorously? 

Reviewer #1: Yes

3. Have the authors made all data underlying the findings in their manuscript fully available?

Reviewer #1: Yes

4. Is the manuscript presented in an intelligible fashion and written in standard English?

Reviewer #1: Yes

5. Review Comments to the Author

Reviewer #1: This is a cross-sectional study on the coverage and utilization of LLINs in a community in Ethiopia. The study reports that although the coverage with LLINs was high, the rate of net utilization was poor. Low utilization was associated with the rural, uneducated households with a lack of awareness and having misconceptions about malaria. Other factors that influenced the rate of utilization included age, the shape of the net, type of beds, housing conditions, employment of the head of house-hold.

The main weakness of this manuscript I that it does not mention the previous exposure of this population to Social and Behaviour Change (SBC) which is generally recommended to be launched with LLINs campaigns. It is interesting to note that Table 3 shows that people who had information about malaria were not better than those who didn’t. This means that if there were SBC programmes, they were not effective. It is important to know about SBC exposure because the conclusion of this study is that this population needs health education. Any future SBC should be evidence-based and could benefit form the findings of this study.

Minor revisions:

(1) There is confusion in using the term “employee” in Table 3 and in the text. Do you mean government employee?

(2) It is contradictory when you attribute the better awareness of employees to better education and at the same time the data do not show correlation with the level of education, clarify this.

(3) Page 8 Paragraph3: households those = households that

6. PLOS authors have the option to publish the peer review history of their article (what does this mean?). If published, this will include your full peer review and any attached files.

Reviewer #1: **Yes: **Ahmed Adeel

---

## [Author Response · Author response to Decision Letter 0]

25 Sep 2023

We appreciate the editor and reviewers’ comments. The following are our point-by-point responses:

Response: Thank you for your comments. This is well received and amended accordingly.

Response: Thank you. We will upload it to public repository recommended by PLOS ONE upon acceptance of the manuscript.

3. We note that Figure 1 in your submission contain copyrighted images. All PLOS content is published under the Creative Commons Attribution License (CC BY 4.0), which means that the manuscript, images, and Supporting Information files will be freely available online, and any third party is permitted to access, download, copy, distribute, and use these materials in any way, even commercially, with proper attribution. 

Response: Thank you for your comments. We took all the images during our data collection to indicate the inappropriate use of LLINs and the images are taken solely by the authors. However, we removed the images that contain (for example the name of the bed net production or distribution company or image that has similarity with possible items on the internet). However, if the editors/reviewers recommend taking out the images regardless, we are happy to incorporate your suggestions.

Response: Thanks for your comments. We have addressed some typos in the reference list and this is indicated in the manuscript accordingly.

Reviewer 2

Comment: Abstract and the background chapter are well written. A Few comments on the methods;

Methods: ”……. The sample size was calculated by using single population proportion formula using proportion of LLINs utilization of 61.6% from similar study” … Please include this refence

Response: Thank you for your feedback. We really appreciate it.The reference is included in the manuscript accordingly and we would also like to apologize and indicate that there is a typo here as the sample size should be 60.6% not 61.6%. Therefore we updated it and included the reference: ‘’ The sample size was calculated by using single population proportion formula using proportion of LLINs utilization of 60.6% from similar study (9).’’

Comment: “A structured questionnaire was administered to identify socio-demographic, LLINs availability and status, LLIN use, and observation was made to inspection condition and handling of the LLINs” there are multiple factors that can potentially affect LLINs utilization, did this tool capture exhaustively capture all these factors/variables? Would it be good idea to indicate how this could have affected your stud and if possible discuss this as a limitation?

Response: We would like to thank the reviewers for the comment. We have tried to assess the socio-demographic, LLINs availability and status factors that can potentially affect LLINs utilization. However, we agree with the reviewer that there would be more known and unkown factors and interrelation between the factors might affect the LLINs utilization resulting in a bias. Therefore, the comment is well received and included in the limitation section.

Comment: “A binary logistic regression analysis was used to examine the determinants of LLIN utilization and variables with a p-value below 0.2 during bivariate analysis was entered into the multivariable logistic regression model” ……. Why was this cutoff set at 0.2? doesn’t setting a much “relaxed”/ less stringent cut off at variable selection potentially retain many unnecessary variables progressing to the final models which essentially could have been dropped

Response: Thank you for the comment. The reason we chose this cut off point is based on (Maldonado and Greenland, 1993) which is recommended to be a reasonable cut off point to rule out confounders more effectively and retain relevant variables for a final model. We have also observed this being used by similar researchers as rule of thumb.

Comment: Results “The median age of the respondent was 37.8 years…” Whats the age mostly affected by malaria in Ethiopia? Is it the under 5?

Response: Thank you for your feedback. The reported age is the median age of the survey respondents or head of the household. As you mentioned, according to the recent systematic review and meta-analysis ( Biset, G., Tadess, A.W., Tegegne, K.D. et al., 2022), the highest malaria-related morbidity and mortality in Ethiopia is reported among children under 5 years. However, in our study we have assessed if children are given priority to sleep under LLINs and found only very few (10.8%) households give priority for children.

Reviewer 1

The main weakness of this manuscript I that it does not mention the previous exposure of this population to Social and Behaviour Change (SBC) which is generally recommended to be launched with LLINs campaigns. It is interesting to note that Table 3 shows that people who had information about malaria were not better than those who didn’t. This means that if there were SBC programmes, they were not effective. It is important to know about SBC exposure because the conclusion of this study is that this population needs health education. Any future SBC should be evidence-based and could benefit form the findings of this study.

Response: Thank you so much. We really appreciate the reviewer’s comment. We agree that it would be great if we have assessed the exposure of the population to Social and Behaviour Change programs. Therefore, we included this in the limitation section of the manuscript as follows: ‘’Also, the findings should be interpreted by taking into consideration that the study did not assess the prior level of community’s exposure to Social and Behaviour Change programs.’’

Comment: Minor revisions: (1) There is confusion in using the term “employee” in Table 3 and in the text. Do you mean government employee?

Response: Thank you. This is well received and amended in the manuscript as ‘government employee’

Comment: (2) It is contradictory when you attribute the better awareness of employees to better education and at the same time the data do not show correlation with the level of education, clarify this.

Response: Thank you and sorry for the confusion here. We wanted to explain that the possible reason is that employees might have an exposure to information about LLIN and malaria and this is clarified in the manuscript accordingly.

Comment: (3) Page 8 Paragraph3: households those = households that

Response: Thank you very much. Comment well received and addressed.

---

## [Editor Report · Decision Letter 1]

18 Oct 2023

Assessment of Long-Lasting Insecticide Nets Coverage, Utilization, And Associated Factors Among Households in Malaria Elimination Districts of Arsi Zone, Oromia Region, Ethiopia

PONE-D-23-19534R1

Dear Dr. Hailu,

We’re pleased to inform you that your manuscript has been judged scientifically suitable for publication and will be formally accepted for publication once it meets all outstanding technical requirements.

Kind regards,

Bosco Bekiita Agaba, PhD

Academic Editor

PLOS ONE
---

## [Editor Report · Acceptance letter]

23 Oct 2023

PONE-D-23-19534R1 

Assessment of Long-Lasting Insecticide Nets Coverage, Utilization, And Associated Factors Among Households in Malaria Elimination Districts of Arsi Zone, Oromia Region, Ethiopia: a cross-sectional study 

Dear Dr. Hailu:

I'm pleased to inform you that your manuscript has been deemed suitable for publication in PLOS ONE. Congratulations! Your manuscript is now with our production department. 

Kind regards, 

on behalf of

Dr. Bosco Bekiita Agaba 

Academic Editor

PLOS ONE